# Sexual health knowledge acquisition processes among very young adolescent girls in rural Malawi: Implications for sexual and reproductive health programs

Wanangwa Chimwaza-Manda[1,2]*, Mphatso Kamndaya[3], Effie Kondwani Chipeta[2], Yandisa Sikweyiya[1,4]

**1** School of Public Health, University of the Witwatersrand, Johannesburg, South Africa, **2** School of Public Health, Kamuzu University of Health Sciences, Blantyre, Malawi, **3** School of Applied Sciences, Malawi University of Business and Applied Sciences, Blantyre, Malawi, **4** Gender and Health Research Unit, South African Medical Research Council, Pretoria, South Africa

* wmanda@cartafrica.org

**Data Availability Statement:** The dataset has been uploaded as a supporting information file.

## Abstract

Early adolescence is a period characterized by enormous biological, cognitive, sexual, emotional, and social changes. Sexual curiosity and the desire to acquire sexual health (SH) information are part of these developments. Understanding the SH knowledge acquisition process is critical for designing interventions that can best support very young adolescents (VYAs). This study explored the SH knowledge acquisition processes among VYA girls aged 10 to 14 years who attended the DREAMs Girl Only Clubs (GOCs) and those who did not. The GOCs were a part of a larger comprehensive HIV prevention project called DREAMS (**D**etermined, **R**esilient, **E**mpowered, **A**IDS-free, **M**entored, and **S**afe) which provided an evidence-based core package of interventions to VYAs to prevent HIV. In-depth interviews were conducted with 43 VYA girls aged 10–14 years in two rural southern districts of Zomba and Machinga in Malawi. Twenty-three VYA girls were GOC participants and 20 VYAs were not. Guided by the Social Ecological Model, a thematic analysis approach was used to analyze the data with the assistance of Nvivo 12 software. The SH knowledge acquisition processes were the interaction of various factors at the microsystem (self-efficacy, attitudes, trust and the beginning of menstruation), mesosystem (communication of SH issues between VYAs and their immediate family and peers), and exosystem levels (availability of life skills programs and mother-groups in schools and availability of GOCs). Compared to Non-GOC participants, GOC participants demonstrated an in-depth knowledge of SH issues and positive sexual behaviors such as HIV testing. Limitations to SH knowledge acquisition were adult messages' focus on sexual relationship avoidance and on girls who have started menstruation; the perception of adults not being knowledgeable about SH and school teachers hiding some SH information. VYAs' SH interventions that provide VYAs with accurate, consistent, and age-appropriate SH information such as the GOCs have the potential to address the limitations that other sources have in reaching VYAs. Integrating such interventions with programs that empower parents, other adults, and teachers with

**Funding:** Funding for the research was provided in part by the generous support of the American people through the United States President's Emergency Plan for AIDS Relief (PEPFAR) and the United States Agency for International Development (USAID) under Project SOAR (Cooperative Agreement AID–OAA–A–14–00060). The contents of this manuscript are the sole responsibility of the authors and do not necessarily reflect the views of PEPFAR, USAID, or the United States Government. In addition, this research was supported by the Consortium for Advanced Research Training in Africa (CARTA). CARTA is jointly led by the African Population and Health Research Center and the University of the Witwatersrand and funded by the Carnegie Corporation of New York (Grant No. G-19-57145), Sida (Grant No:54100113), Uppsala Monitoring Center, Norwegian Agency for Development Cooperation (Norad), and by the Wellcome Trust [reference no. 107768/Z/15/Z] and the UK Foreign, Commonwealth & Development Office, with support from the Developing Excellence in Leadership, Training, and Science in Africa (DELTAS Africa) programme. The statements made and views expressed are solely the responsibility of the author. The funders had no role in study design, data collection, and analysis, the decision to publish, or the preparation of the manuscript. There was no additional external funding received for this study.

**Competing interests:** The authors declare that they have no competing interests

comprehensive SH information and with skills on how to deliver SH information to VYAs can enhance VYAs' SH knowledge acquisition and influence positive behavior change.

## Introduction

Early adolescence is defined as the period between ages 10 and 14 years. This period, designating the transition from childhood to adulthood is considered crucial because it establishes the basis for future sexual practices and gender beliefs [1]. During this period, adolescents undergo tremendous biological, cognitive, sexual, emotional, and social changes [1,2]. Early adolescence is also characterized by sexual curiosity, whereby most young adolescents develop a desire to learn about sexuality [3]. Although the majority of VYAs have not yet engaged in penetrative sex, some have reportedly engaged in foreplay, heavy petting, kissing, and other non-coital acts [4–6].

As the VYAs undergo various developmental changes, they also acquire information [2]. Literature suggests that early adolescence is also a period where misinformation regarding SH issues is prevalent from various erratic sources of information [1,7,8]. Exposure to misinformation predisposes VYAs to SH challenges such as early unintended pregnancies and STI infections, including HIV [9]. Previous studies have shown that VYAs can be protected from SH challenges if they receive age-appropriate, accurate and consistent information [10–12]. Studies conducted globally have shown that parents, other adults, peers, and social institutions within the social contexts in which adolescents live are the main sources of information for SH [13–15]. However, few studies have explored SH knowledge acquisition processes among VYAs.

As the advocacy for the inclusion of VYAs in SH interventions is increasing, there is need to ensure that SH interventions that target VYAs equip them with SH information that is accurate, consistent, and age-appropriate, in order to meet the unique SH needs and challenges of VYAs [2,12]. Exploring knowledge acquisition processes is one way of ensuring that SH interventions are tailored to VYAs' SH needs [16,17]. Also, exploring the SH knowledge acquisition processes among VYAs could provide insights into their preferred sources of information, the most effective methods of knowledge acquisition, and the barriers they encounter in accessing information, leading to the provision of SH information that is accessible and engaging for VYAs [16,18]. Furthermore, exploring knowledge acquisition processes could lead to the identification of gaps in SH knowledge among VYAs. Such information could enable programmers to provide targeted, accurate, and comprehensive information that addresses these gaps and capacitate or support VYAs to make informed decisions regarding their SH [16,18].

Literature suggests that the process of acquiring SH knowledge is influenced by cultural norms, values, and contextual factors [16,17,19,20]. Through an understanding of these factors, practitioners may be able to design SH interventions for use with VYAs that are culturally sensitive and contextually relevant. Furthermore, such an understanding will enable the programmers to develop SH interventions that consider the diverse beliefs and practices of the VYAs while still providing accurate and evidence-based SH information [16,17,19,20].

In Malawi, like in most countries, the majority of VYAs rely on schools, parents, peers, and the media for their SH knowledge [12,21]. VYAs also acquire SH information from a few adolescent health programs, such as in-school Life Skills and Comprehensive Sexuality Education programs [22,23]. Understanding how VYAs interact with these sources of information and how they acquire SH information from these sources is critical for developing SH interventions that are suitable for VYAs.

**Table 1. Classification of Go Girl Club DREAMS interventions.**

| | The DREAMS GO! Girl Club Intervention Classification | | |
|---|---|---|---|
| | **10–14 years** | **15–19 years** | **20–24 years** |
| **Primary Individual Interventions** | •Social Asset Building<br>•HIV Testing Services<br>•Condom information<br>•Screen for case management | •Social Asset Building<br>•HIV Testing services<br>•Condoms<br>•Access to Contraceptive Information and Services | •Social Asset Building<br>•HIV Testing Services<br>•Condoms<br>•Access to Contraceptive Information and Services |
| **Secondary Individual Interventions** | •Combination of Socio-economic Approaches (VSL) for caregivers<br>•Food Security/ Nutrition<br>•Post Violence Care<br>•Access to Contraceptive Information and Services<br>•Back to School Support | •Combination of Socio-economic Approaches (VSL)<br>•Food Security/ Nutrition<br>•Post Violence Care<br>•Back to School Support<br>•Parenting Skills<br>•Case Management<br>•Recency Test | Combination Socio-economic Approaches (VSL)<br>Food Security/ Nutrition<br>Post Violence Care<br>Back to School Support<br>Parenting Skills<br>•Recency Test |

The Go Girl Club initiative is an example of an SH intervention that reached VYA girls with SH information in two southern rural districts, Zomba and Machinga, in Malawi. The two districts were chosen by the **D**etermined, **R**esilient, **Empowered, A**IDS-Free, **M**entored, and **S**afe Lives (DREAMS) Initiative implementers because of their high HIV prevalence (13% and 16.3% among 15-49-year-olds, in Zomba and Machinga, respectively), high proportions of orphans and vulnerable children below the age of 18 (20.7% and 18.9%, in Machinga and Zomba, respectively), high prevalence of early sexual initiation (20.1% and 24.1% in Machinga and Zomba, respectively), high rates of childbearing during adolescence (26.1% and 31.6% among the 15-19-year-olds in Machinga and Zomba, respectively), and high primary school drop-out rates among girls (17.3% and 12.6% in Machinga and Zomba, respectively) [24,25]. In the DREAMS initiative, the VYA girls were included in health interventions such as sexual and reproductive health (SRH)-related skills-based training, including HIV prevention, education assistance, social skills, asset development, and economic strengthening through the Girl Only Clubs (GOCs), called the Go Girl Clubs.

The GOCs were established in 2015 as part of the DREAMS initiative, which was funded by the US Government's President's Emergency Plan for AIDS Relief (PEPFAR) [26]. The goal was to cut the number of new HIV infections among adolescent girls and young women (AGYW) by 40% in 15 countries, including Malawi. To recruit the AGYW into the GOCs, messages were sent to the villages by the implementing organization through community structures such as traditional leaders to invite all AGYW between the ages of 10 and 24 to participate in the clubs (Table 1). Those who expressed interest to participate were recruited into the clubs. The clubs were categorized according to the following age groups: the 10–14-year-old clubs; the 15–19-year-old clubs; and the 20–24-year-old clubs. Each GOC comprised 10 to 25 participants. The GOC participants met in the clubs once a week. In these clubs, young women aged between 18 and 24 called mentor mothers used the My Dreams, My Choice tool kit, which is a form of a curriculum, to train the participants (AGYW) in various skills relating to their SH [27]. The skills include building social assets, learning about their body, communication skills, their future and vision, and STIs and HIV [25,28]. In addition, the parents of the VYAs were trained in positive parenting skills.

Findings from an implementation science study on the GOCs initiative's impact on AGYW aged 15 to 24 years have shown that there was an increase in comprehensive knowledge about HIV and condoms, and higher levels of support for equitable gender norms as well as higher relationship power among the AGYWs who went through the intervention [25,29]. However,

the study did not explore the knowledge acquisition process among the VYA girls (aged 10–14 years) and the impact of GOCs on SH outcomes among this age group. The current qualitative study aimed to fill this gap by exploring the SH knowledge acquisition processes among VYA girls who attended the DREAMs GOCs and those who did not. The inclusion of a comparison group of VYAs who did not attend GOCs created a basis for comparison to examine whether club participation contributes to differences in SH knowledge acquisition among VYAs. The knowledge acquisition process is defined as the gathering and processing of information, the application of information to problems and tasks, and the anticipation of how current actions affect future outcomes [30]. In this study, the SH knowledge acquisition process is defined as the gathering of SH information from various sources, including how VYAs perceived these sources, and the factors influencing the gathering of SH information.

We used Bronfenbrenner's Social Ecological Model (SEM) to understand the processes of SH knowledge acquisition among VYA girls [31–34]. The model considers a person's (for example, a VYA girl's) growth in the context of the relationships that make up her environment [32]. Bronfenbrenner divided the environment into levels in which the person interacts. The microsystem, which is the environment closest to the individual and contains psychological qualities, individual characteristics, and behaviors (e.g. individual attitudes and self-efficacy) is the first level [35]. The second level is the mesosystem which includes the family, relational, and community spheres (e.g., neighborhoods, and schools) [35]. The third level is the exosystem which includes characteristics of the society at large (e.g., socio-economic status, healthcare policies, media, and gender) [35]. The fourth category is the macrosystem which comprises cultural values, customs, and laws [36–38]. Bronfenbrenner's SEM has been used in studies to understand adolescents' health by identifying determinants of their health located at the individual level as well as those located in the community, structural and socio-physical environments [39–42]. The assumption behind using Bronfenbrenner's SEM in the current study is that the SH knowledge acquisition processes among the VYA girls will be influenced by various factors at the different levels of the SEM.

## Methods and materials

This was a qualitative exploratory study that explored the SH knowledge acquisition processes among VYA girls who attended the DREAMs GOCs and those who did not.

### Study setting

The study was conducted in two locations in Zomba. The first was a catchment area of 63 villages where 88 GOCs existed, and where the DREAMS initiative was being implemented. VYAs were present in only two of the 88 GOCs, with a total of 20 VYA girls. The second site had no DREAMS activities and was about 60 kilometers away from where DREAMS activities were taking place.

In Machinga, two sites were chosen as well. The initial site was where the DREAMs project activities were implemented, and it had a catchment area of 45 villages with 75 GOCs. There was just one GOC with VYA girls (n = 10), The second site was purposefully chosen as there were no DREAMS activities present, and the site was about 50 kilometers away from the first site.

### Sampling and data collection

Purposive sampling was used to recruit participants for this study [43]. The participants were divided into two groups. The first group comprised VYA girls in the DREAMS sites who were aged 10 to 14. VYA girls in the DREAMS sites had been members of the club for about a year

at the time of the interviews. To participate in the study, the VYA girls were recruited through an invitation that was shared with them with the help of a person working for the DREAMS project. We approached 30 VYA girls between the ages of 10 and 14 to participate in the study; however, we could only include VYA girls between the ages of 12 and 14 who met our inclusion criteria, which included availability and willingness to participate, as well as the ability to communicate experiences and opinions in an articulate, expressive, and reflective manner. To assess their ability to communicate and express themselves, the VYAs were asked some informal ice-breaking questions (e.g., what they do in their free time and why, who their best friends are, and why they perceive those people as their best friends) before the informed consent procedures. Data were collected between 2018 and 2019 for a period of two months. A total of 23 in-depth interviews (IDIs) were conducted in Machinga and Zomba (S1 Appendix). As noted, in Machinga, there was only one VYAs GOC with a total of 10 VYA girls. In this GOC, seven VYA girls were interviewed; three club members refused to be interviewed after expressing their discomfort with the interviews. In Zomba, there were two VYAs GOCs with a total of 10 members each. Eight interviews with VYA girls were done in each of Zomba's two clubs. On the day of the interviews, two club members from each of the two clubs were unavailable.

The second group comprised VYA girls from villages outside of the DREAMS implementation area. To our knowledge, there had been no HIV or SH programming for VYAs in the area around the time we conducted the interviews. The District Youth Officer found community-based Youth Volunteers who aided in the identification of VYA girls who were eligible for the study. VYA girls (aged 10–14) who had never been to a youth club were approached and asked to take part in the study. A total of 20 VYA girls were interviewed (S2 Appendix). Data saturation was reached after conducting 10 interviews at each site.

The interviews were conducted in October 2019 by the first author, who is a female qualitative researcher with prior experience working with adolescents. Interviews were done in Chichewa, a dominant spoken language in the study setting. The interviews were conducted face-to-face before the onset of the COVID-19 pandemic. To ensure privacy and confidentiality, a private house was arranged within the community where individual interviews took place. The decision to conduct the interviews at a private house was because there were concerns around privacy in the homes of the VYAs. It was reasoned that the lack of privacy may make the respondents not speak freely and in a candidate manner if they thought that someone else could overhear what they were saying. Also, there was a concern that at the homes of the VYAs there might be siblings or friends that will either want to listen in on the interview out of curiosity or keep interrupting the respondent for other reasons.

The VYA girls' SH knowledge acquisition processes were investigated using a semi-structured interview guide (S1 Appendix and S2 Appendix). Knowledge of HIV/AIDS, STIs, pregnancy and contraception, and menstruation were the four areas of SH knowledge investigated. The VYA girls were asked to narrate stories about the first time they heard about HIV/AIDs, STIs, pregnancy, contraception, and menstruation respectively. The question was followed up with a series of probes about where they first heard of these issues, who talked to them about these issues, whom they asked and why they chose to ask those people, what triggered their curiosity about these issues and what they did with the information shared with them. All the interviews were one-time events. Each interview was 30–40 minutes long.

## Ethical considerations

Ethical approval for this study was obtained from the College of Medicine Research Ethics Committee (COMREC) in Malawi (Ethics Approval Number: PP.01/17/2095), the University

of the Witwatersrand Human Research Ethics Committee (HREC) in South Africa (Approval number: M181009), and the Population Council in the United States of America (Approval number: 784). To ensure that the study was done ethically and that the data collection techniques protected the participants and reduced potential harm, guidelines for interviewing children and adolescents were followed [44]. Before engaging the VYA girls, their parents or guardians were approached to obtain their permission to interview the VYAs, and those who agreed signed an informed consent form. After obtaining the VYA girl's parents/guardian's consent, VYA girls gave their assent to participate in the study.

## Reflexivity

The findings presented in this article were drawn from a qualitative study that employed a narrative inquiry approach and was nested in the DREAMs implementation science study. The DREAMS implementation science study assessed the DREAMS initiative implementation in Zomba and Machinga districts in Malawi. The first author who was a study coordinator for the DREAMS implementation science study was involved in the protocol development, development of data collection tools, data collection, and analysis of the qualitative data.

For the nested qualitative study, the first author conducted all the in-depth interviews with the VYA girls. Employing a narrative inquiry approach, the first author explored VYA girls' experiences of SH knowledge acquisition. The narrative inquiry approach enabled the VYAs to freely share their thoughts, opinions, and experiences with the first author through a narration of their life stories. The first author's positionality as an adult researcher influenced the choice of the narrative inquiry as she deemed it a friendly and appropriate approach for conducting qualitative interviews with very young adolescents.

Considering the ages [10–14 years] of the study participants, being interviewed by an adult researcher may have created a power imbalance during the interviews. The first author was an older woman and the VYA girls may have thought of her as someone who required particular answers from them. Alternatively, out of respect that the first author was an older woman, the VYA girls might have provided her with responses that they felt were socially desirable. To reduce this potential bias, we conceptualized the interview as a dialogue and conversation and wrote the interview questions in an open-ended manner. Also, the first author made a conscious effort to establish rapport with the participants and to create a comfortable, non-judgmental environment for the interviews. Throughout the interviews, the first author reminded the participants that they had a right to withdraw from the study at any time if they did not feel comfortable continuing. In conducting the interviews in storytelling manner, the first author experienced respondents as comfortable expressing their thoughts, and opinions and in sharing their life experiences. However, there was a difference between GOC participants and Non-GOC participants in terms of how they expressed themselves about SH issues. This was not very surprising as we had expected that GOC participants would be more open and informed about SH issues because of their exposure to SH information due to club participation.

## Data analysis

Audio-recorded IDIs were transcribed verbatim in preparation for data analysis. The transcripts were then painstakingly translated from Chichewa into English while maintaining the original meanings. A research assistant was employed to handle the transcribing and translation. The first author read the transcripts and compared them to the respective audio to ensure that they were accurate. Following that, the transcripts were imported into the NVIVO program (NVIVO12, QSR International).

To analyze the data, we employed a thematic analysis approach [45]. To acquaint herself with the text, the first author read all transcripts several times. Following further readings of five transcripts, the first codes reflecting significant topics in the transcripts were created to form a codebook. Another independent qualitative researcher was engaged in coding to test the codebook and assess whether there was inter-coder agreement [43,46].

Relationships between the codes were explored, and codes with similar elements were grouped into main themes. For instance, all texts illustrating the knowledge acquisition processes about HIV formed the first main theme, all texts illustrating knowledge acquisition processes about pregnancy issues formed a second theme, and all texts illustrating knowledge acquisition processes about menstruation issues formed the third theme. We compared the experiences of GOC and Non-GOC participants using a constant comparative analysis (CCA) approach. The CCA is an analytic method that entails a set of systematic procedures relating to assigning codes or categorizing data and subsequently identifying themes or patterns [47,48]. To conduct the CCA all data were first analyzed separately (i.e., for VYA girls in GOCs and VYA girls not in Clubs). Secondly, the themes within and between the two different groups of participants were compared. Further analysis was done to examine the knowledge acquisition processes using the levels of the SEM which include the microsystem, mesosystem, exosystem, and macrosystem levels. The data for VYA girls in GOCs and that for VYA girls not in clubs were first analyzed separately, thereafter the themes within and between the two groups were compared.

## Results and discussion

### Social demographic characteristics of study participants

A total of 43 VYA girls participated in the interviews (Table 2). The age of the respondents ranged between 10 and 14, with the youngest aged 10–11 among the non-GOC participants. The majority of the respondents were in primary school (78% GOC participants, 80% Non-GOC participants), with none in secondary school. Most of the GOC participants lived with both parents while most of the non-GOC participants lived with relatives at the time of the interviews.

**Table 2. Social demographic characteristics of participants.**

| Characteristic | GOC Participant | | Non-GOC Participants | |
|---|---|---|---|---|
| | Number (N = 23) | % | Number (N = 20) | % |
| **Age** | | | | |
| 10 | 0 | 0 | 1 | 5 |
| 11 | 0 | 0 | 2 | 10 |
| 12 | 5 | 22 | 4 | 20 |
| 13 | 9 | 39 | 5 | 25 |
| 14 | 9 | 39 | 8 | 40 |
| **Education Status** | | | | |
| In school (Primary school) | 18 | 78 | 16 | 80 |
| Out of school | 5 | 22 | 4 | 20 |
| **Housing (living with)** | | | | |
| Both Parents | 15 | 65 | 6 | 30 |
| Single Parent | 5 | 22 | 4 | 20 |
| Other relatives | 3 | 13 | 10 | 50 |

## Knowledge of SH issues

We explored VYA girls' knowledge of SH issues including HIV/AIDS, pregnancy and contraception, and menstruation. Specifically, we investigated the VYA girls' general awareness of HIV/AIDS and how its transmitted, how unwanted pregnancy can be prevented, their knowledge of the types of contraceptives and their understanding of menstruation and its onset. We further examined how the participants learned about the above issues and how they make their decisions regarding which source of support to use or not to use to address their SH needs. We found that most GOC and Non-GOC participants were aware of these issues. For instance, most respondents were able to describe what HIV/AIDs is; how HIV/AIDS could be acquired; and how HIV could be prevented.

> "*HIV is sexually transmitted, if someone does not want to contract the disease, has to use condoms*" [**GOC Participant, Age 12, Machinga**]

> "*HIV/AIDS is a disease that is transmitted through unprotected sexual intercourse, Sharing razor blades, needles, and toothbrushes*" [**Non-GOC Participant, Age 12, Machinga**]

Furthermore, GOC participants and non-GOC participants indicated some knowledge about pregnancy and how it could be prevented:

> "*I heard about what happens when one gets pregnant but one cannot get pregnant when she has sex using a condom, if the sperms enter your body you might get pregnant but if they don't enter your body you cannot get pregnant. But about what happens when one gets pregnant I don't know.*" [**Non-GOC Participant, Age 14, Machinga**]

> "*. . .to avoid pregnancy one has to use contraceptives and use condoms when having sex*" [**GOC Participant, Age 13, Machinga**]

For both GOC and Non-GOC participants, knowledge about menstruation varied between VYA girls who had already started menstruation at the time of the interview and those who had not started menstruation. Most respondents who had experienced menses reported that they first knew about menstruation when they experienced it for the first time:

> "*Yeah, by the time I was developing my breasts and I also started experiencing menstruation then I was really surprised such that I consulted elders to counsel me on this. I was really afraid about it because I had never experienced that, so I wanted to seek guidance about this.*" [**GOC Participant, Age 13, Zomba**]

> "*And about menstruation (the other name is monthly periods), I did not know. Me, I learned it from the book. And when I saw it happening to me, I stayed for two days; afraid, saying "should I tell them? Or whom should I tell? Won't they say that I've gotten hurt or feel embarrassed?" I tried in one way or another but I failed to take care of myself. I was discovered and then people advised me as to how it is.*" [**Non-GOC Participant, Age 14, Machinga**]

In contrast, those who had not started menstruation said they did not know much about menstruation, some said they had never heard about it, and others said they overheard people talking about it.

**Interviewer**: Have you experienced your first period?

**Respondent:** No.

**Interviewer**: Have you ever discussed menstruation with anyone?

**Respondent**: No. I just heard it from my sisters, they were discussing it, and I just over-heard it.

**Interviewer**: What questions do you have about menstruation?

**Respondent:** I do have questions like what happens for menstruation to start and also, how to stop it.

[GOC Participant, Age 13, Machinga]

Among the GOC participants, some respondents mentioned that they first heard about menstruation at the GOCs.

**Interviewer:** Mmmh, okay, what else, I also wanted to know the benefits you have gained from the Go Girl club as compared to before you joined the club.

**Respondent:** I have benefited lots of things as I was blank in various things. . . Yeah, I have benefited a lot because I wasn't aware of menstruation.

[GOC Participant, Age 12, Machinga]

## Knowledge acquisition processes

In this article, we define the SH knowledge acquisition process as the gathering of SH information from various sources, how VYAs interact with these sources, and the factors that influence how they gather this information [30]. Both GOC and Non-GOC participants identified parents, older adults, peers, and school as the main sources of SH information. However, in addition to this, GOC participants identified the GOCs as another source of SH information. VYA girls said they acquired SH information from various sources, and we found that several factors at the individual, interpersonal and institutional levels affected SH knowledge acquisition processes among VYAs (see Table 3).

## Individual level factors

Individual-level factors refer to the characteristics, behaviors, and attributes of individual people that influence their interactions, choices, and outcomes within a particular context or

**Table 3. Summary of SH knowledge acquisition processes.**

| Socio-ecological model level | Themes | Knowledge acquisition processes | Sources |
|---|---|---|---|
| Individual level | Self-efficacy to seek SH information<br>Trust in the sources of Information<br>Attitude toward the sources of information | Seeking SH information from others<br>Receiving SH information from others | Parents, older relatives, Friends |
| Interpersonal level | Parents/older relatives' initiative to communicate SH issues<br>Inquiry and discussion of SH Issues with friends | Communication of SH issues by parents or relatives<br>Inquiry from peers, discussions with peers | Parents, older relatives, Friends |
| Community level | Availability of Life Skills Programs in schools<br>Availability of Mother groups in schools<br>Availability of GOCs | Learning through the Life skills subject in school, counseling from mother groups, learning through GOC | Schools, GOCs |

environment [35]. In this study, individual-level factors refer to attributes of the VYA girls that influenced their SH knowledge acquisition such as self-efficacy, trust, and attitude.

*Self-efficacy to seek SH information.* Self-efficacy refers to an individual's belief in their ability to successfully perform specific tasks, attain goals, or handle challenging situations [49]. In this study, self-efficacy refers to the VYAs belief or confidence to seek SH information. In the interviews, participants were asked whom they talk to when they have questions about SH issues. Some VYA girls said they approached their parents or older relatives for SH information. Specifically, the participants who indicated that they had a parent or older relation whom they felt was supportive of them were more likely to report that they approached such a parent or an older relative for SH information:

"*My mother told me that if I want to ask I should not fear her, I can ask her or other people, so I ask my mother about these [SH] other things.*" [**Non-GOC Participant, Age 13, Zomba**]

"*I ask my in-law about these things whom I talk with about these girly things. She tells me about her background. That when she was a girl she liked being in sexual relationships but still the relationships were just for fun. She says it is just to tell us that somehow getting involved in so many sexual relationships; is not good. I am also able to tell her when I am maybe having my monthly period.*" [**GOC Participant, Age 14, Machinga**]

*Trust in the sources of information.* Some VYA girls said they preferred to talk about SH issues with older people whom they trust for confidentiality. For instance, like others, the participant below said she preferred to talk with her mother, instead of a friend, because of trust issues:

"*If it was with my mom, no one would spread the word about what we talked about [SH issues], but if it was with a friend, they might tell someone else what we discussed.*" [**non-GOC Participant, Age 12, Zomba**]

On the other hand, some participants said they prefer to talk to their parents and other adults because they thought that the SH information they received from their peers could be misleading:

"*. . . because girls say rudely that it is better to contract syphilis other than HIV, if one gets syphilis, eh things will be bad for her than HIV, but we don't talk about such issues very well.*" [**Non-GOC Participant, Age 13, Machinga**]

Some participants, however, preferred not to talk to parents because they felt parents were not well informed about SH issues:

"*Parents from this community, they cannot explain things that we learn at school, sometimes they can and sometimes they also cannot explain, and that makes you opt not to ask again since you know they will not explain.*" [**Non-GOC Participant, Age 13, Machinga**]

*Attitude towards sources of information.* Several participants indicated that they had never spoken or discussed SH issues with their parents. They gave several reasons for this, including fearing their parents' reaction, and their experiences that their parents were not willing to discuss such issues. Moreover, other participants felt it was not proper to talk about SH issues, with their fathers:

*"Umm, my father I fear him. And, he says "I cannot talk more to you, right? it is not my part."* *So, our Dads do not talk to us much about HIV."* [**GOC Participant, Age 12, Zomba**]

*"I have never discussed HIV with my father because am a girl and he is a man so this is not good."* [**GOC Participant, Age 14, Zomba**]

Other participants intimated that they could not talk about SH issues because they felt shy:

*"No, I have never discussed or asked questions about HIV with my parents. I just feel shy to do."* [**GOC Participant, Age 13, Machinga**]

*The beginning of menstruation.* The data suggest that for both GOC participants and non-GOC participants their parents or older relatives started to talk to them about the prevention of HIV and pregnancy, and contraception after the VYA girls had experienced their first menstruation. Most participants said when their parents or older relatives became aware that they had started menstruation they told them that they should keep away from boys as they may impregnate them or infect them with HIV:

*"When it started [menstruating], I told my mother. She taught me to wear sanitary pads during periods, she told me that, don't walk around unnecessarily, and don't have sexual intercourse with boys. If you have sexual intercourse with boys, you will get pregnant"* [**GOC Participant, Age 13, Machinga**]

*"When it [menstruation] happened, I was at my mother's sister's place. I was getting up and I saw my skirt stained with blood and knowing that I do not have a wound, I thought of telling her. She told me that you are now grown up and that I need sanitary pads and that she would buy them for me, and she explained to me how to wear them. She told me that I need to take care now because if I start having boyfriends and having sex I can get pregnant or HIV"* [**non-GOC Participant, Age 14, Zomba**]

## Interpersonal level factors

These refer to the interactions, relationships, and social dynamics between individuals and the people around them. In this study, interpersonal level factors refer to the communication about SH issues between VYAs and their immediate family and peers.

*Parents/Older relatives' initiative to communicate SH issues.* The data suggest several pathways through which VYA girls would acquire SH knowledge or information from parents. Firstly, some participants acquired SH information through a parent's or older relative's initiative to give SH advice to the VYA girls. This initiative occurred through a parent or older relative having an impulse to advise a VYA girl, when they suspected that a VYA girl was in a sexual relationship, and when a VYA girl had started menstruation. Secondly, some participants reported that they acquired SH information through their initiative to approach parents or older relatives.

Participants who reported parents or relatives as sources of SH information were probed further on how they learned about SH issues from parents or older relatives. Several participants reported that parents or older relatives advised them randomly to discourage or prohibit them from engaging in romantic relationships. Such messages came as a form of warning about negative SH outcomes such as contracting HIV or STI or even becoming pregnant if they engaged in a sexual relationship:

"*My parents said that I should not have sex with anyone or get into a relationship so that I may not contract diseases. . .My father said that he should never see me being in a relationship with a boy, but I should work hard on my education*" [**GOC Participant, Age 13, Zomba**]

"*My parents were talking saying that, 'out there, there is HIV, my child, a disease. . ..out there, there is HIV, it is a disease, so you have to be careful.' So, I said 'all right.'*" [**GOC Participant, Age 12, Machinga**]

Also, when asked about how they learned about SH issues from parents, other participants reported that parents or older relatives advised them about SH issues when they suspected or noticed that they had a boyfriend. This advice also focused on telling the VYA girls how they should protect themselves from negative SH outcomes in sexual relationships. Apart from receiving prohibitive messages against engaging in sexual relationships, some participants said they received advice from their parents or older relatives on how to prevent pregnancy or contracting HIV:

"*My grandmother said that to avoid pregnancy one has to use contraceptives and use condoms when having sex. . .I was chatting with a boy, and she came and started advising me that boys can make me contract diseases and that my friends use condoms to avoid pregnancy or diseases and others take contraceptives.*" [**GOC Participant, Age 14, Machinga**]

**Interviewer**: All right. What about your Auntie, as you said. Have you ever discussed issues concerning HIV and AIDS?

**Respondent**: Yes.

**Interviewer**: You were discussing based on what? What were you discussing?

**Respondent**: She asked me "do you have a boyfriend?" . . . I answered her "yes." she said "but be careful, there are sexually transmitted diseases like HIV and AIDS. So, to prevent it, you can use condoms." So, I responded, "Okay." "And also, you need to go and get tested, you should know your status." So we got together and went with her.

[GOC Participant, Age 14, Machinga]

*Inquiry and discussion of SH Issues with friends.* Several respondents also reported that they learn about SH issues from friends. This was reported by both GOC participants and non-GOC participants. Our data suggest that VYA girls discuss SH issues amongst themselves including sexual relationships, menstruation, HIV/AIDS, pregnancy, and contraception:

"*I heard that if the sperms enter your body you might get pregnant but if they don't enter your body you cannot get pregnant. But about what happens when one gets pregnant I didn't know. I asked my friends how one would be noticed if she is pregnant. That is when they told me that some are seen because of change in complexion, sometimes you get fat, some elders would ask you to show them your palms, and sometimes you don't do your period and that is how I knew.*" [**Non-GOC Participant, Age 13, Machinga**]

Our data show that the VYA girls also advise each other about SH and warn each other about negative SH outcomes:

"*I first heard about HIV from my friends. They were saying that HIV can be contracted through sexual intercourse, sharing of sharp objects like razor blades, needles and using the same toothbrush with infected person. . .*" [**GOC Participant, Age 13, Machinga**]

*"My friend. She told me that "don't have unprotected sex with other people because you could get pregnant and your mother doesn't have the resources to take care of you if you do get pregnant." Someone else also told me not to have unprotected sex with people older than me. I then realized that If I get pregnant my mother wouldn't have the resources to take care of me.".* [**Non-GOC Participant, Age 13, Machinga**]

Unlike messages from parents which mainly focused on warning against sexual relationships or on how to prevent negative SH outcomes, our data suggest that the SH information that VYA girls discussed with friends went beyond prevention of SH outcomes. For instance, VYAs girls discussed amongst themselves issues such as their experiences in romantic relationships:

*"We also discuss issues of relationships with boys, like for her, she had had a boyfriend before, and we discuss such things. She once had a boyfriend, and she was telling me that there are problems in relationships, I asked her what problems, and she told me that relationships are troublesome and that I should not try them. She said that people hurt each other. . ."* [**Non-GOC Participant, Age 14, Zomba**]

*"My friends were just telling me. They say they are given money and enough love by their boyfriends"* [**Non-GOC Participant, Age 13, Machinga**]

On the other hand, some participants they felt that the SH information they received from their peers could be misleading:

*". . . because girls say rudely that it is better to contract syphilis other than HIV, if one gets syphilis, eh things will be bad for her than HIV, but we don't talk about such issues very well."* [**Non-GOC Participant, Age 13, Machinga**]

### Community level factors

Community-level factors in the socioecological model refer to the characteristics, and dynamics, including the availability and accessibility of resources within a community, such as schools, healthcare facilities, parks, community centers, libraries, and social services, that can significantly impact residents' quality of life. In this study, both GOC participants and non-GOC participants reported that they also acquired SH information from the school. Two pathways were reported. First, some participants said they acquired SH knowledge through learning about it at school, and through the mother group clubs, located in the schools. Second, GOC participants said the GOCs were their additional source of SH information.

*Availability of life skills programs in schools.* Participants who reported acquiring SH information from school said they learned about SH issues in class through the Life skills subject which is part of the school curriculum:

*"I heard about these issues for the first time at school during life skills. They said that a person is supposed to avoid getting diseases. . .you should avoid getting pregnant. A person should not often be having sexual relationships because you can get pregnant when you did not plan for it. If you do not avoid getting pregnant it might result in you failing to attain your education goals and in so doing it means you have failed to achieve your future dreams." [GOC Participant, Age 13, Zomba]*

The participants also said the messages delivered in class also focused on STI and pregnancy prevention and that the SH information acquired from schools focused on clarifying concepts such as HIV and AIDS:

"*She (the teacher at school) said that HIV is a virus that causes diseases and people contract HIV through sexual intercourse and sharing toothbrushes. AIDS is when one begins to suffer from many ailments due to HIV.*" [**Non-GOC Participant, Age 14, Zomba**]

While the school was reported to be a source of SH information, some participants said some SH information gets hidden from them at school by their teachers:

**Interviewer**: Have you ever heard or been taught about such issues at school?

**Respondent**: Yes, one-time female teachers called us all girls at school, and they explained that.

**Interviewer**: What about during life skills subject?

**Respondent**: They do explain but some teachers explain explicitly and others hide some facts.

**Interviewer**: Why do you think they hide some things?

**Respondent**: I don't know but some teachers explain things properly while others just cut them through and in that way, you may not understand things, but others explain very well from the roots to the end while others do not.

**Interviewer**: Do they allow you to ask questions when they finish teaching?

**Respondent**: Yes, but others who don't explain things when you try to ask them that you didn't understand they tell you, that you will understand when you are in heaven.

[Non-GOC Participant, Age 14, Machinga]

*Availability of school-based Mother-groups*. The data suggest that VYA girls also acquired SH information through school-based mother group clubs. The mother group clubs comprised volunteer women from the villages who came to the school to teach VYA girls about SH issues including menstruation issues [50–52]. However, the participants reported that the mother groups targeted only girls who had started menstruation:

"*We have a mother group at our school. They call all menstruating girls in a class then they tell us that, they explain how to avoid pregnancies.*" [**GOC Participant, Age 13, Zomba**]

"*At school, there are other Madams, where they call up all girls who have started having periods to give the same advice. For example, telling us that when we go to watch games we should not go home late to avoid us getting pregnant. They pick up older girls only; they do this as mother groups.*" [**Non-GOC Participant, Age 12, Machinga**]

*Availability of GOCs*. GOCs participants reported the clubs as another source of SH information, with several participants indicating that they had heard for the first time about HIV, other STIs, pregnancy, and contraception at the GOC:

"*I started going to the club in September. I joined because of my friends, right? A lot of them were saying "let's go and join, let's join." So, I said "I will not join, because I do not know what*

*I will learn there. I stopped school a long time ago. So, I should be going there, what will I learn? And they said, "go, go" so I went. And when I went, it was when we were learning about pregnancies; how can we avoid pregnancies or HIV how can we avoid it. We learned it there at One Community for the first time."* [**GOC Participant, Age 13, Zomba**]

*"In this village, I think it is the first time to have an organization [GOC] that is teaching girls about AIDS, I have never heard about any organization coming and teaching HIV and AIDS. I have benefitted because I have learned how to avoid AIDS, and I have known how a girl's body changes something I did not know but now to me it's now not a new thing anymore. And I have learned how to avoid unplanned pregnancies and a lot more."* [**GOC Participant, Age 14, Machinga**]

**Differences between GOC and Non-GOC participants in SH knowledge acquisition.** Compared to SH information acquired from parents or older adults, friends, and schools, the SH information received from clubs seemed to be comprehensive. Our data suggest a difference between GOC participants and non-GOC participants in terms of knowledge of contraceptive methods. For example, the non-GOC participants mentioned only the condom as a contraceptive method whereas the GOC participants mentioned a variety of contraceptive methods in addition to condoms:

*"At the clubs, we have learned how to take care of our bodies and avoid pregnancies, such as how to use condoms, injections, or loops."* [**GOC Participant, Age 13, Zomba**]

While all participants regardless of whether they were GOC participants or not, reported learning about protecting themselves from contracting STIs and from getting pregnant using condoms, the GOC participants reported learning how to use a male condom through demonstrations done in the clubs:

*"So we learned about how to wear a condom; everyone when coming; maybe they choose people, right? Our facilitator would say to you 'when coming to the club, you should bring a penis-looking object; maybe some bring a maize cob' and others will bring a bottle. Then they will dress those objects for demonstration."* [**GOC Participant, Age 13, Machinga**]

Furthermore, GOC participants showed great retention of SH information they acquired from the clubs. This was demonstrated by the detailed SH information GOC participants shared when they narrated what they learned in clubs.

*"I had questions like how to protect ourselves and the use of condoms and its side effects. So I learned from one community (GOCs). I wanted to know how I could protect myself from pregnancy and how to use a condom and I wanted to know what happens when the condom has torn apart. So, I heard all about these issues at the club, and they answered me. I have benefited lots of things as I was blank in various things."* [**GOC Participant, Age 13, Machinga**]

*"I have benefited a lot from learning at the clubs. For example, during our monthly periods we did not understand many things, right? But we understood them through the book we used to learn from at the clubs. During the monthly periods, it is important to take care of yourself So we realized that we were not properly taking care of ourselves in the private parts during monthly periods. But after we learned from this book at the club, we can take care of ourselves."* [**GOC Participant, Age 14, Machinga**]

On the other hand, several non-GOC participants showed gaps in knowledge when they were asked to mention things they know about SH issues including pregnancy and HIV:

"*So I have been asking myself how people get pregnant. . . because there was this month I missed my menstruation. Yes, because people say that if you miss a month without menstruating then you are pregnant, so I was wondering I haven't slept with a man, but I have missed my menstruation, so I have been thinking then I just saw that the other month I did on the first date of the month.*" [**Non-GOC Participant, Age 12, Machinga**]

"*Mmm, I have them to say this HIV if I happen to be found with it, what would I do? I ask myself how this HIV, how does it come? Yeah, I ask myself such questions.*" [**Non-GOC Participant, Age 13, Zomba**]

Another notable difference in the SH information acquired from the GOCs was that the GOC participants were provided with information on where to go for HIV testing. For instance, some GOC participants reported going for an HIV test which was something they attributed to joining the clubs:

"*It happened that when we were just joining, they told us that we should go and get tested for HIV; the same people from One Community (GOCs). So, the first benefit I found was that one.*" [**GOC Participant, 13, Zomba**]

"*At that time, they (GOC facilitators) told us that if one wants to get tested, she should go to the hospital and that there are some personnel from the club so if you make an appointment with the club facilitator you will just have to meet that person.*" [**GOC Participant, 14, Machinga**]

Compared to the teachers at school where some participants said school teachers often hid 'sensitive' SH information, several GOC participants lauded GOC facilitators for their commitment to teaching them about SH issues:

"*I like learning from our club teacher, I should say that our teacher at the club teaches us with a committed heart because they are not paid, they just teach us.*" [**GOC Participant, 13, Zomba**]

"*About sexuality issues, I prefer speaking with our club facilitator, because she teaches us these things and therefore knows better than our friends or parents. . .*" [**GOC Participant, 14, Machinga**]

## Discussion

This study explored the SH knowledge acquisition processes among VYAs girls who attended the DREAMs GOCs and those who did not. Broadly, our findings show that SH knowledge acquisition processes for DREAMs GOC participants and non-GOC participants are almost similar. Also, our findings are consistent with those of previous studies which have shown that the main sources of SH knowledge among VYAs are parents, older relatives, friends, and school [1,9,14,53,54]. Using the SEM as an analytical framework, we found that the SH knowledge acquisition processes among the VYA girls were influenced by the interaction of various factors in the microsystem (biological factors, attitudes), mesosystem (parents, older relatives, friends), and exosystem levels (schools and GOCs).

At the microsystem level, the individual attitudes of the VYA girls including the trust and the confidence they had in a potential source of SH information influenced where they sought SH information. For instance, some participants indicated they would rather speak with an adult than with a friend, and this stemmed from their view that an adult can keep things confidential than friends. Yet, some participants indicated that their parents were not fully equipped with SH knowledge, therefore they preferred to seek SH information elsewhere. Our findings suggest that GOC participants trusted their club facilitators as sources of comprehensive SH information. Taken together, these findings accentuate the need to empower VYAs' parents and older relatives with comprehensive SH information and skills on how they can appropriately deliver SH information to VYAs. Moreover, they suggest the need for alternative sources of SH information such as GOCs to be made available to VYAs so they can comfortably access the information.

At the mesosystem level, we found that parents and older relatives were one of the main sources of SH information for VYA girls. Yet, our finding that most parents/older relatives only start to talk to VYA girls about SH issues (e.g., prevention of pregnancy) when they find out that the girl has started menstruation is concerning. This approach does not only exclude VYA girls who have not yet begun menstruation, and deprive them of access to SH information, advice, and guidance from their parents and older relatives but also places them at risk of negative SH outcomes such as HIV, STIs, and pregnancy. This is especially true given that most VYAs are sexually aware, and some have started engaging in non-penetrative sexual activities like kissing, fondling foreplay, and heavy caressing [3,4,6,55].

We have presented a finding showing that some VYA girls were more comfortable seeking SH information from parents or older relations whom they felt supported them on SH issues. Our findings are similar to those of previous studies which have shown that young girls who perceived their parents as open and comfortable to talk about sexual issues were more inclined to approach them for SH knowledge, guidance, and advice [14,56]. This is a positive shift regarding parent-child communication about sexuality issues in SSA because previous studies have reported social-cultural taboos hindering parent-child communication on sexuality issues [57]. This finding suggests the importance of including a component of strengthening the quality of parent-child communication in SH interventions that target VYAs [13,58].

We have also presented a finding showing that both GOC participants and non-GOC participants felt that the SH messages given by parents and older relatives tended to focus on telling VYA girls to avoid sexual relationships. This finding may have implications for VYAs especially those who may be already sexually active at the point they get this message [59]. For example, a VYA girl who is sexually active and seeking knowledge on how to protect herself from a negative SH consequence is unlikely to approach her parents or elder relatives if they had previously discouraged sexual relationships. This finding is similar to that of a study conducted among VYAs in Kenya where adolescents were reluctant to reveal their sexual or romantic connections because they were afraid of negative parental reactions if they disclosed their engagement or interest in romantic partnerships [59].

The friends of VYA girls were described as another source of SH information at the mesosystem level, and this was true for both GOC participants and non-GOC participants. However, we have shown that while some SH information acquired from friends was accurate, the VYAs also received misinformation from their friends. This is in line with the research that shows that in early adolescence, friends, siblings, and other sources of information abound with misinformation about sexuality concerns [1,7,9,60]. This finding underscores the need for VYAs to have alternative reliable child-friendly places or forums within their reach where they could access correct SH information.

At the exosystem level, we found that the school and the GOCs were the main sources of information. However, there were differences in terms of the influence of the school and the GOCs on SH knowledge cognition. For instance, we found that the VYA girls who participated in GOCs had indicated more in-depth knowledge of SH relative to their counterparts who were non-GOC participants. Furthermore, we presented a finding which suggests that SH information received from school was not always complete as some teachers reportedly hid some SH information from them. Teachers withholding or being uncomfortable delivering explicit SH information to students has been reported in previous studies in SSA [61–63]. This finding suggests a gap in skills among teachers in primary school on how to deliver SH information to VYAs. In our study sample, 22% of GOC participants and 20% of Non-GOC participants were out-of-school girls. Thus, the finding that GOCs were one of the main sources of SH information is also especially significant for out-of-school girls who may not have access to in-school sex education programs.

In summary, our findings indicate that despite that the VYAs have various sources of SH information within their communities, these sources have some critical challenges or limitations. These challenges or limitations need to be addressed through evidence-based programs to capacitate parents or adults with SH knowledge and skills on how to engage constructively with VYAs on SH issues. Our findings suggest that the GOCs play an important role in addressing some of the limitations that were found with the other sources of SH information (e.g., parents, older relatives, friends, and schools). For example, while most parents and older relatives were reported to only start talking about SH issues after the VYA girls start menstruation, the GOCs were said to target all VYAs regardless of whether they had started menstruation or not. Furthermore, the GOCs were experienced by the GOC participants as providing comprehensive SH information and practical guidance. The provision of accurate and age-appropriate knowledge and skills through the GOCs enabled the VYAs to not only protect themselves from acquiring STIs but also make informed choices regarding their SH. In addition, whereas the life skills program at schools reaches only in-school VYA girls, the GOCs reach both in-school and out-of-school VYA girls. Last, unlike the other sources of information, our findings indicate that the GOCs provided a neutral and safe environment where VYA girls can go and seek SH information.

## Implications for sexual and reproductive health programming for VYAs

Our research has several points of significance on adolescent SRH programming in Malawi and similar settings. First, our findings support the view that adolescent SRH interventions should target adolescents in their early adolescent years rather than in late adolescence [64]. Targeting adolescents at an early age equips them with accurate knowledge and skills including communication, negotiation, decision-making, problem-solving, and assertiveness [1,6,10,65]. For young adolescents, these skills are essential for building a strong foundation for healthy relationships and responsible sexual behavior in adulthood [64]. Our findings have highlighted the need for reliable sources of SH information for VYA girls. Therefore, SRH programming or interventions should include empowering adults, parents, teachers, religious leaders, and other community members with comprehensive SRH information and empowering them on how to deliver this to VYAs in families, schools, and communities. Specifically, SRH programming should target to promote positive parenting skills and positive dialogue and communication between parents and VYAs. Our findings have also highlighted the need for SRH programmers to create safe and supportive spaces within communities where VYA can easily access SRH information. The GOCs in Malawi are an example of such programs that involve social asset building and safe spaces, especially for VYAs, and have been demonstrated to indirectly reduce the risk for

HIV because they increase agency and empowerment among AGYW [64,66]. Finally, our findings have also highlighted the significance of insuring that SH programs that target VYAs provide them with SH information that is accurate, consistent, and age-appropriate.

## Strengths and limitations of the study

The study has several strengths. Firstly, the use of the SEM provided us with a framework for understanding how multiple and interacting factors affected the SH knowledge acquisition processes among the VYA girls. Secondly, the inclusion of a comparison group of Non-GOC participants added to the study's strength as it allowed us to compare the influence of the GOCs on SH knowledge acquisition processes among the VYAs.

There are some limitations to this study. Firstly, each respondent had only one interview. Each interview took between 30 and 45 minutes to complete. Secondly, the age of the respondents had an impact on the total amount of time spent with each of them, and because of their young age, most of them did not go into much detail in their comments. Fourthly, the age gap between the researcher and the VYAs who took part in the study could have limited how much they stated. All these factors may have limited the time in which the respondents had to say more and the time the interviewer had to delve further. However, the use of narrative inquiry which entailed that participants respond to questions in a storytelling manner, helped to address some of these challenges as this approach enabled the establishment of rapport with the VYAs by allowing them to be in control of how they tell their stories and to focus on the issues that they felt were worth talking about. Another limitation is that we did not look at the perspectives of other stakeholders, such as club facilitators, parents, and other family relations to the club participants. Future studies would do well to expand the scope to obtain the perspectives of other stakeholders, as this would enable the studies to gain a holistic understanding of the SS dynamics and their influence on the SH needs of VYA girls. In addition, we did not explore the role of mass media in knowledge acquisition. This limited our understanding of the influence of or the role mass media may play on knowledge accuracy and quality. We therefore recommend that future studies should consider exploring the role of mass media in SH knowledge acquisition processes among VYAs. Furthermore, we only compared the SH knowledge acquisition processes between the VYAs who were in school and those out of school. This study would have been strengthened if we had older adolescents as a comparison group. Comparing the knowledge acquisition processes between VYAs and older adolescents would have provided a more comprehensive understanding of the factors influencing SH knowledge acquisition including the developmental differences, peer influence, decision-making processes, and risk perceptions that may be unique to the two groups of adolescents. Having such data would have enabled the development of tailored interventions for each age group. Lastly, the SEM was useful in exploring how multiple and interacting factors affected the SH knowledge acquisition processes among the VYA girls, however, as learning and knowledge acquisition is a complex process and involves cognition, it was limited in exploring cognitive processes among VYAs. Future studies should thus consider integrating the SEM with the social cognitive theory as this would allow for a more comprehensive understanding of knowledge acquisition processes among adolescents by examining individual cognitive processes through the use of social cognitive theory within broader social and environmental contexts through the use of the SEM.

## Conclusion

This study investigated the SH knowledge acquisition processes among the GOC and Non-GOC participants. Our study highlights the limitations that sources of SH information such as parents, relatives, friends, and schools have in reaching VYA girls. However, the findings of

this study have also shown that avenues such as the GOCs have the potential to address the limitations that parents, relatives, friends, and schools have in reaching VYAs with SH information. Thus we argue that SH interventions targeted at VYAs should include empowering their parents, other adults, and teachers with comprehensive SH information and skills on how to deliver SH information to VYAs. Lastly, we contend that there is need to increase the avenues for VYAs in rural Malawi where they could acquire accurate, consistent, and age-appropriate information about SH issues. This is crucial for protecting VYAs from negative SRH outcomes.

## Supporting information

**S1 File.**
(ZIP)

**S1 Appendix. English IDI Guide for girls in clubs.**
(DOCX)

**S2 Appendix. English IDI Guide girls not in clubs.**
(DOCX)

## Acknowledgments

The authors would like to thank the DREAMS Implementation Science research team, including colleagues at the Population Council and Centre for Reproductive Health at the University of Malawi, College of Medicine, who made this work possible. Special thanks to One Community for facilitating access to their program participants and for sharing their time and expertise. Thanks to all the girls who participated in the study who committed their time and shared their experiences and opinions.

## Author Contributions

**Conceptualization:** Wanangwa Chimwaza-Manda.

**Formal analysis:** Wanangwa Chimwaza-Manda, Yandisa Sikweyiya.

**Funding acquisition:** Wanangwa Chimwaza-Manda.

**Investigation:** Wanangwa Chimwaza-Manda.

**Methodology:** Wanangwa Chimwaza-Manda, Effie Kondwani Chipeta.

**Supervision:** Mphatso Kamndaya, Effie Kondwani Chipeta, Yandisa Sikweyiya.

**Validation:** Mphatso Kamndaya, Effie Kondwani Chipeta.

**Visualization:** Yandisa Sikweyiya.

**Writing – original draft:** Wanangwa Chimwaza-Manda, Mphatso Kamndaya, Effie Kondwani Chipeta, Yandisa Sikweyiya.

**Writing – review & editing:** Wanangwa Chimwaza-Manda, Mphatso Kamndaya, Effie Kondwani Chipeta, Yandisa Sikweyiya.

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
