## [Decision Letter · Decision Letter 0]

27 Apr 2023

PONE-D-22-27569Sexual health knowledge acquisition processes among very young adolescent girls in rural Malawi: Implications for sexual and reproductive health programsPLOS ONE

Dear Ms. Manda,

Thank you for submitting your manuscript to PLOS ONE. After careful consideration, we feel that it has merit but does not fully meet PLOS ONE’s publication criteria as it currently stands. Therefore, we invite you to submit a revised version of the manuscript that addresses the points raised during the review process.

We look forward to receiving your revised manuscript.

Kind regards,

Catherine Aicken

Guest Editor

PLOS ONE

Journal Requirements:

 “Funding for the study was provided in part by the generous support of the American people through the United States President’s Emergency Plan for AIDS Relief (PEPFAR) and the United States Agency for International Development (USAID) under Project SOAR (Cooperative Agreement AID–OAA–A–14–00060). The contents of this manuscript are the sole responsibility of the authors and do not necessarily reflect the views of PEPFAR, USAID, or the United States Government.

In addition, this study was supported by the Consortium for Advanced Research Training in Africa (CARTA). CARTA is jointly led by the African Population and Health Research Center and the University of the Witwatersrand and funded by the Carnegie Corporation of New York (Grant No--B 8606.R02), Sida (Grant No:54100029), the DELTAS Africa Initiative (Grant No: 107768/Z/15/Z). The DELTAS Africa Initiative is an independent funding scheme of the African Academy of Sciences (AAS)’s Alliance for Accelerating Excellence in Science in Africa (AESA) and supported by the New Partnership for Africa’s Development Planning and Coordinating Agency (NEPAD Agency) with funding from the Well-come Trust (UK) and the UK government.” 

Additional Editor Comments (if provided):

Thank you for submitting your manuscript. The reviewers acknowledge the strengths of the manuscript - which I also recognise - but have asked for some revisions. I believe these will help you to improve the paper.

When you provide your revised manuscript, please respond to each of the reviewers' points - either explaining how and where you have made the change(s), or providing a rebuttal. Regarding reviewer 2's comment 3, I do agree that it would be helpful to include a reflexive account but not necessarily within the Methods. On this point, you may choose to briefly describe the interviewers'/researchers' positionality in the Methods and add a slightly longer (e.g. one paragraph) reflexive account in the Discussion.

Regarding the statement about data availability, please confirm the reasons why the original data are not shared (e.g. participant confidentiality and lack of consent to share transcripts), unless you have done so already.

I am looking forward to reading your revised manuscript.

Reviewers' comments:

Reviewer's Responses to Questions

**Comments to the Author**

1. Is the manuscript technically sound, and do the data support the conclusions?

Reviewer #1: Yes

Reviewer #2: Yes

2. Has the statistical analysis been performed appropriately and rigorously? 

Reviewer #1: N/A

Reviewer #2: N/A

3. Have the authors made all data underlying the findings in their manuscript fully available?

Reviewer #1: No

Reviewer #2: No

4. Is the manuscript presented in an intelligible fashion and written in standard English?

Reviewer #1: Yes

Reviewer #2: Yes

5. Review Comments to the Author

Reviewer #1: Thank you for the opportunity to review this manuscript. It presents an important piece of work that addresses relevant public health issues around improving sexual health knowledge among very young adolescents for behaviour change. Generally, the manuscript is well-written however, I found a few major challenges with the approaches used and the presentation of the data. I outline below some of the concerns and specific challenges:

1. It is imperative to indicate in the introduction and through-out the manuscript the importance of not only providing ‘appropriate’ information – but rather ‘accurate, consistent and age-appropriate information’. This would tie in with their findings around misinformation and partial-information which is not consistent.

2. Although the intervention Girls Club/ Go Girl Club initiative is nested/embedded within the DREAMS initiative, a box or section describing it in detail would’ve helped put the intervention in context of the findings and discussion. A separate sub-section to describe or reference what the GOC was about, who delivered it, how often, where would be useful. Also, include the ages it targeted and how it recruited its participants would be useful information for the manuscript reader. [page 5 line 77-83]

3. Line 89. What was the justification of focusing on knowledge acquisition processes on VYA, was that what the intervention was about? Would not having older adolescents as a comparative group been a better approach maybe to shed more light on why it is important to understand knowledge acquisition processes in the VYA?

4. Line 93. The SEM is a good broad framework to use in general. However, for this study the Social Cognitive Learning Theory would’ve probably been better to show the interaction between the individual and the various layers of the SEM that the authors present but acknowledge how environment influences behaviour (The Social Cognitive Learning Theory acknowledges the constant interaction that exists between the individual and his or her environment, both structural and social, to shape behaviour).

5. Line 105-108. Following from above comment, I found the SEM a bit too simplistic. Learning and knowledge acquisition is complex process and involves cognition, self-efficacy, attitude etc.

6. Study setting line 114-121. It would be great if the authors indicate how the two districts were selected and add a few statistics to show the magnitude of HIV, HIV treatment gaps, OVC, early sexual initiation and school-drop-out rates.

7. Line 130-145. It would be interesting to elaborate how these VYAs were approached for the study and how the assessment of ability to participate was conducted, considering that some expressed discomfort with the interviews and unavailability of participants. Also, in the discussion, the interviewers were indicated to have been a limitation – were they trained in conducting interviews with children, could the study have used other child friendly approaches to engage the children attention for longer and capture data rather than long interviews? Ranking, sorting and participatory methods, etc could’ve been employed.

8. When was the study conducted and for long?

9. Line 157-159. The interviews were conducted in a private home in the community, where all the participants taken to this place? Why were the interviews not conducted within the children’s homes? I understand parental consent to participate in the GOC and interviews was obtained. I understand this was a sensitive study and may disclose VYA sexual behaviour to the parents – but if parental consent was obtained, they would’ve known what the study was about and maybe VYA would’ve been comfortable doing interviews in their homes? Please clarify

10. Results: Line 10 – I think the proportion of VYA not in school is worth noting in the discussion especially as the authors indicate schools were a source of info and VYA who were not in the GOC would’ve probably not received SH in schools.

11. Line 269-326. I found the findings quite repetitive and could’ve been easily combined. This is mostly a presentation issue and probably trying to frame the results around the SEM as no new themes emerged/were generated at different layers.

12. Line 293 – this sentence is not clear – please look and clarify.

13. Line 371 -379 – the quotes seem repeated – were there other quotes from the other VYA which can be drawn from?

14. Line 441 – if you could please explain what the “school-based mother group clubs” are and how they are different from GOC, what they teach? How did they identify girls who would’ve started menstruating?

15. Line 618 – I would probably add/rephrase the sentence so it is a more positive approach and ensure that knowledge and skills go beyond just protecting themselves but making informed decisions/choices regarding their sexual health.

16. Line 636-644. For each limitation listed, it would be great to show why/how it was a limitation and the show how the study went around he issues highlighted where possible.

Reviewer #2: Review report

Title: Sexual health knowledge acquisition processes among very young adolescent girls in rural Malawi: Implications for sexual and reproductive health programs Manuscript Number: PONE-D-22-27569

Dear Editor

Thank you for the opportunity to review this work. The paper addresses an important public health issue on sexual and reproductive health knowledge acquisition processes and sources of information for the very young adolescent girls in Malawi. The authors present an in-depth interview analysis of VYA from intervention and non-intervention sites and analyse thematically. It is a well-written paper and envisage to show how knowledge is acquired by VYAs. The authors employed the social ecological model to depict the multilevel and contextual aspects of knowledge acquisition. However, the paper needs to address some issues to improve the presentation of the result and readability:

Major Comments:

Abstract and background

1. The explanation in the study setting section line 114-119 that gives the statement of the problem should be moved to the background section as it provides robust contextual information.

“This study was carried out in Malawi's rural southern districts of Zomba and Machinga. Youth aged 10 to 24 make up one-third of the population in both districts (40). The districts were chosen because of their high HIV prevalence (13% and 16.3% among 15–49-year-olds, in Zomba and Machinga, respectively), HIV treatment gaps, high proportions of orphans and vulnerable children, high prevalence of early sexual initiation, high rates of childbearing during adolescence, and high school drop-out rates among girls (26).”

2. The rationale and the aim of the research need to be clearly articulated to connect the background with methodology.

Methodology

3. Can you dedicate a section in the methodology for a reflective account of your role in the DREAMS project, on the process of the interview, on the reaction of the VYAs to your questions, on what your expectations between the GOC and Non-GOC was, and how that changed during the data collection period and beyond?

Result

In my opinion this is where your manuscript needs more work to clearly answer your research question and attend to the complexity of the phenomenon you are investigating:

4. What knowledge did you examine -- e.g., overall awareness, knowledge on mode of transmission, perceived susceptibility, or knowledge on prevention? This needs to be clearly stated in the paper.

5. Though your primary research objective is to explore the knowledge acquisition process, the themes seem limited to the sources of SRH knowledge (parent, peers, school, clubs). I would expect a deeper examination of the knowledge acquisition processes (communication, learning, mentoring, attitudes, competence on verification of reliability of information etc) and how they are different between the SEM levels. Also, the multiplicity of the sources is not emphasized e.g., one VYA’s could have multiple sources of information and acquire knowledge through different processes.

6. I see you have indicated in the limitation how interview with other stakeholders could have contributed to the model, however the use of SEM still should be justified by presenting the levels, how the various levels interact, what factors influenced VYAs knowledge acquisitions and behaviours? A summary table that shows the SEM levels and the themes, factors, knowledge acquisition processes, the sources of information at each level would be helpful.

Line 631-633 “The study has several strengths. Firstly, the use of the SEM provided us with a framework for understanding how multiple and interacting factors affected the SH knowledge acquisition processes among the VYA girls.”

I don’t see the multiple and interacting factors that affected the SH knowledge acquisition process?

7. Was it possible to conduct a comparison between the VYAs from the implementation sites and VYAs outside the implementation areas? Was there a difference in a source (apart from the clubs, did they communicate better with their parents and peers), was there a difference in the knowledge acquisition process, factors influencing the knowledge acquisition process, was the quality of knowledge different? Line 197 that themes were compared between the two groups. Could you present them?

8. What role did media and technology play if explored? Do the VYAs have access to media and technology, including mass media (radio and TV), social media etc..?

Minor comment

1. Your abstract is unstructured and a little bit difficult to follow. Can you provide a structured abstract and especially emphasis on the themes and the key findings from your analysis.

2. Can you provide the semi-structured interview guide as a supplementary material?

6. PLOS authors have the option to publish the peer review history of their article (what does this mean?). If published, this will include your full peer review and any attached files.

Reviewer #1: **Yes: **Natsayi Chimbindi

Reviewer #2: No

---

## [Author Response · Author response to Decision Letter 0]

11 Aug 2023

Reviewer 1.

1. It is imperative to indicate in the introduction and through-out the manuscript the importance of not only providing ‘appropriate’ information – but rather ‘accurate, consistent and age-appropriate information’. This would tie in with their findings around misinformation and partial information which is not consistent.

Response: 

Thank you for this important suggestion. We have added a paragraph in the introduction section on the importance of providing accurate, consistent, and age-appropriate information to the third paragraph in the introduction section. Please refer to lines 72 to 88 highlighted in yellow in the introduction section, lines 740 to 742 in the discussion section, and lines 764 to 766 in the Implications for SH programming section

2. Although the intervention Girls Club/ Go Girl Club initiative is nested/embedded within the DREAMS initiative, a box or section describing it in detail would’ve helped put the intervention in context of the findings and discussion. A separate sub-section to describe or reference what the GOC was about, who delivered it, how often, where would be useful. Also, include the ages it targeted and how it recruited its participants would be useful information for the manuscript reader. [page 5 line 77-83]

Response

Thank you for this important suggestion. We have added a thick description of the Go Girl Club initiative including a box describing the interventions for the age groups. Please refer to lines 112 to 125 in the introduction section highlighted in yellow. 

3. Line 89. What was the justification of focusing on knowledge acquisition processes on VYA, was that what the intervention was about? Would not having older adolescents as a comparative group been a better approach maybe to shed more light on why it is important to understand knowledge acquisition processes in the VYA?

Response

Thank you for this important observation. We have provided the justification for focusing on knowledge acquisition processes in this age group. Please refer to lines 75 to 88 highlighted in yellow. In this study, the aim was to assess the specific influence of club participation on sexual health knowledge acquisition among very young adolescent girls, as such we chose a comparison group of very young adolescent girls who did not participate in the clubs. The justification for using a comparison group of VYAs who did not participate in the GOCs has now been included. Please refer to the introduction section lines 132 to 135 highlighted in yellow. We also agree with the reviewer that having older adolescents as a comparison group would be another better approach. Therefore, we have now included a recommendation in the Limitations section that future studies exploring a similar question should consider having older adolescents as a comparison group. Please refer to line number 773-780 highlighted in yellow.

4. Line 93. The SEM is a good broad framework to use in general. However, for this study the Social Cognitive Learning Theory would’ve probably been better to show the interaction between the individual and the various layers of the SEM that the authors present but acknowledge how environment influences behavior (The Social Cognitive Learning Theory acknowledges the constant interaction that exists between the individual and his or her environment, both structural and social, to shape behavior).

Response

Thank you for this very important suggestion. During the analysis we carefully considered both the social-cognitive theory and the social-ecological model, considering both their strengths and limitations. However, we chose to use the social-ecological model, its limitations notwithstanding. 

We agree with the reviewer that the social cognitive theory may have enabled understanding of the cognitive processes that influence knowledge acquisition, however, the social cognitive theory has a limited emphasis on context. Furthermore, the social cognitive theory primarily focuses on individual cognitive processes and their interaction with the environment, through concepts like self-efficacy, observational learning, and self-regulation. However, as mentioned above, it places less emphasis on broader social, cultural, and environmental contextual factors that can significantly impact knowledge acquisition. Additionally, while the social cognitive theory acknowledges the role of social influences, such as modeling and social support, it may not fully capture the complexity of social interactions that occur in the process of knowledge acquisition. Adolescents' learning experiences are shaped by their interactions with peers, teachers, mentors, and other significant individuals. 

Lastly, the social cognitive theory does not extensively address the influence of broader environmental contexts, such as community resources, school settings, and cultural norms. These factors play a crucial role in shaping learning opportunities and determining the availability of resources for knowledge acquisition. By not fully considering these environmental influences, the social cognitive theory may provide an incomplete understanding of the factors that contribute to or hinder knowledge acquisition among adolescents.

5. Line 105-108. Following from above comment, I found the SEM a bit too simplistic. Learning and knowledge acquisition is a complex process and involves cognition, self-efficacy, attitude etc.

Response

Thank you for this important comment. Our view above notwithstanding, we have now described the limitations of the SEM in the Limitations sections. Further to this, we have included a recommendation for future studies to consider an integration of the two theories to have a more holistic exploration of the individual’s cognitive processes within the broader social and environmental contexts. Please refer to lines 779 to 780 in the Strengths and Limitations section. 

6. Study setting line 114-121. It would be great if the authors indicate how the two districts were selected and add a few statistics to show the magnitude of HIV, HIV treatment gaps, OVC, early sexual initiation and school-drop-out rates.

Response

Thank you for this important comment. We have indicated how the districts were selected. We have also added the statistics that were used by the DREAMs Initiative implementers as a justification for implementing the DREAMS initiative in these two districts. The statistics include HIV prevalence (13% and 16.3% among 15–49-year-olds, in Zomba and Machinga, respectively), high proportions of orphans and vulnerable children below the age of 18 (20.7% and 18.9%, in Machinga and Zomba respectively), high prevalence of early sexual initiation (20.1% and 24.1% in Machinga and Zomba respectively), high rates of childbearing during adolescence (26.1% and 31.6% among the 15-19-year-olds in Machinga and Zomba respectively), and high primary school drop-out rates among girls (17.3% and 12.6% in Machinga and Zomba respectively). Please refer to the Methods section, lines 97 to 103. 

7. Line 130-145. It would be interesting to elaborate how these VYAs were approached for the study and how the assessment of ability to participate was conducted, considering that some expressed discomfort with the interviews and unavailability of participants. Also, in the discussion, the interviewers were indicated to have been a limitation – were they trained in conducting interviews with children, could the study have used other child friendly approaches to engage the children attention for longer and capture data rather than long interviews? Ranking, sorting and participatory methods, etc could’ve been employed.

Response

We thank the reviewer for this important question. We have now elaborated on the approach that was used to recruit the participants (i.e., VYAs) and how their ability to participate in the study was assessed. Please refer to the sampling section, lines 176-177. And also lines 182 and 184. Regarding the limitation on interviewers: We have now clarified that all the interviews were conducted by the first author. We have now also explained that the age gap between the researcher (1st author) and the respondents, and their age could have limited what they said in the interviews. The interviewer (first author) is a seasoned qualitative researcher who has extensive experience working and doing research with children. Moreover, we used the narrative inquiry approach, which is a participatory research method that uses a storytelling approach. This approach is deemed to be a friendly and appropriate approach for collecting qualitative data from children[1–3]. For instance, the story telling nature of a narrative inquiry helps in rapport building which allows children to tell stories in their own way and to focus on key issues that are important to them. Please refer to the Reflexivity section line number 241-252. Please also refer to the Strengths and Limitations section line number 754 to 763.

8. When was the study conducted and for long?

Response

Thank you for this important question. The data collection for the study was conducted between 2018 and 2019 for a period of two months. We have included this information in the sampling and data collection section. Please refer to lines 184-185.

9. Line 157-159. The interviews were conducted in a private home in the community, where all the participants taken to this place? Why were the interviews not conducted within the children’s homes? I understand parental consent to participate in the GOC and interviews was obtained. I understand this was a sensitive study and may disclose VYA sexual behavior to the parents – but if parental consent was obtained, they would’ve known what the study was about and maybe VYA would’ve been comfortable doing interviews in their homes? Please clarify.

Response

Thank you for this important question. The decision to conduct the interviews at a private house was because we had concerns around privacy in their homes which would have possibly made the respondents to not speak freely and in a candidate manner when they knew that someone else could potentially overhear what they were saying. Also, we were concerned that at the children’s homes, siblings or friends might either want to listen in on the interview out of curiosity or keep interrupting the respondent for other reasons. As such, we decided that interviews will be conducted in a private place within the community. This was also a preferred approach by the Research Ethics Committees that approved this study. We have now included this information in the manuscript. Please refer to the sampling and data collection section, lines 204 to 209.

10. Results: Line 610 – I think the proportion of VYA not in school is worth noting in the discussion especially as the authors indicate schools were a source of info and VYA who were not in the GOC would’ve probably not received SH in schools.

Response

Thank you for this important observation and suggestion. We have now included the proportion of VYAs who were not in school in the discussion section. Please refer to the discussion section lines number 708 to 709 highlighted in yellow.

11. Line 269-326. I found the findings quite repetitive and could’ve been easily combined. This is mostly a presentation issue and probably trying to frame the results around the SEM as no new themes emerged/were generated at different layers.

Response

Thank you for this important comment. We agree that the data presented in lines 269-326 may seem repetitive. However, these data speak to three instances in which parents communicate to VYAs about SH issues. The first instance relates to how parents may provide random advice about SH issues to the VYAs. The second instance relates to communication and advice that was given to VYAs because the parent was suspicious that the VYA was engaging in a romantic relationship. The third instance shows advice that came from parents because the VYA girl was known to have started her menstruation. Please refer to line number 436 to 448, 449 to 469, and 409-417. 

12. Line 293 – this sentence is not clear – please look and clarify.

Response

Thank you for this important observation. We have revised the sentence. Please refer to line number 449-451. 

13. Line 371 -379 – the quotes seem repeated – were there other quotes from the other VYA which can be drawn from?

Response

Thank you for this important comment. We have revisited the quotes referred to and noted that they are two different quotes. The quote which was in line 371 (now 475 to 480) illustrates a discussion among VYAs which focused on describing the signs of a pregnant woman. The quote in line 379 (now 483-485) reflects a discussion among VYAs regarding the sources of information about HIV and how to prevent HIV. We have thus maintained the quotes as they are. 

14. Line 441 – if you could please explain what the “school-based mother group clubs” are and how they are different from GOC, what they teach? How did they identify girls who would’ve started menstruation?

Response

Thank you for this comment. The GOCs are clubs based in the communities sponsored by the DREAMS initiative as explained in lines 103 to 116 in the introduction section. On the other hand, the mother groups are volunteer women from the community who come to teach adolescent girls about SH and also encourage girls to remain in school. These are not part of the DREAMS initiative. We have added some references where more about mother groups have been explained. Please refer to line number 552 to 553, highlighted in yellow. 

From the quotes in line lines 556 to 562, highlighted in yellow, the girls who started menstruation were identified in school classes where a teacher shared information with the girls above the availability of the mother groups who would be able to offer information and advice to the girls about menstruation. The teachers then encourage the girls who had started menstruation to speak to the mother groups. 

15. Line 618 – I would probably add/rephrase the sentence so it is a more positive approach and ensure that knowledge and skills go beyond just protecting themselves but making informed decisions/choices regarding their sexual health.

Response

Thank you for this important suggestion. We have revised this sentence as per the reviewer’s suggestion. Please refer to line number 710 to 712. 

16. Line 636-644. For each limitation listed, it would be great to show why/how it was a limitation and show how the study went around he issues highlighted where possible.

Response

Thank you for this important suggestion. In line 754 to 759 highlighted in green, we have described the limitations of the study highlighting four factors These factors related to the amount of time that the respondents had during the interview, and the limited time that the interviewer had to explore in-depth the issues and topics explored in this study. In line 760 to 763 highlighted in yellow, we have explained that the use of narrative inquiry which entailed that participants respond to questions in a story telling manner, helped to address some of these limitations. We have explained that the narrative inquiry approach helped in building rapport with the VYAs by allowing them to tell their stories in their own way and to focus on key issues that they felt were important to talk about. 

Response to Reviewer 2’s comments

17. Abstract and background

The explanation in the study setting section line 114-119 that gives the statement of the problem should be moved to the background section as it provides robust contextual information.

“This study was carried out in Malawi's rural southern districts of Zomba and Machinga. Youth aged 10 to 24 make up one-third of the population in both districts (40). The districts were chosen because of their high HIV prevalence (13% and 16.3% among 15–49-year-olds, in Zomba and Machinga, respectively), HIV treatment gaps, high proportions of orphans and vulnerable children, high prevalence of early sexual initiation, high rates of childbearing during adolescence, and high school drop-out rates among girls (26).”

Response

Thank you for this important suggestion. We have now moved this section to the introduction section. Please refer to the introduction section, fifth paragraph line number 97 to 103.

18. The rationale and the aim of the research need to be clearly articulated to connect the background with methodology

Response

Thank you for this important comment. The rationale and the aim have now been clearly articulated to connect background with methodology. Please refer to lines 126 to 132. 

19. Can you dedicate a section in the methodology for a reflective account of your role in the DREAMS project, on the process of the interview, on the reaction of the VYAs to your questions, on what your expectations between the GOC and Non-GOC was, and how that changed during the data collection period and beyond?

Thank you for this important suggestion. We have added a reflexivity section in the methods section. Please refer to line number 228 to 252 highlighted in yellow. 

20. Results

In my opinion this is where your manuscript needs more work to clearly answer your research question and attend to the complexity of the phenomenon you are investigating:

What knowledge did you examine -- e.g., overall awareness, knowledge on mode of transmission, perceived susceptibility, or knowledge on prevention? This needs to be clearly stated in the paper.

Response

Thank you very much for this important comment. We have now added a description of the type of knowledge that we investigated in this study. Please refer to line 290 to 293 in the results section highlighted in yellow. 

21. Though your primary research objective is to explore the knowledge acquisition process, the themes seem limited to the sources of SRH knowledge (parent, peers, school, clubs). I would expect a deeper examination of the knowledge acquisition processes (communication, learning, mentoring, attitudes, competence on verification of reliability of information etc) and how they are different between the SEM levels. Also, the multiplicity of the sources is not emphasized e.g., one VYA’s could have multiple sources of information and acquire knowledge through different processes.

Response

Thank you for this important comment. We agree that the themes presented seem to be limited to the sources of SH knowledge for VYAs which were parents, peers, schools and clubs. To more clearly show the knowledge acquisition processes, we have revised the presentation of the themes by reorganizing into 3 categories in line with the three SEM levels: individual, interpersonal, and community level factors. Please refer to the results section from line number 364 to 366. 

22. I see you have indicated in the limitation how interview with other stakeholders could have contributed to the model, however the use of SEM still should be justified by presenting the levels, how the various levels interact, what factors influenced VYAs knowledge acquisitions and behaviors? A summary table that shows the SEM levels and the themes, factors, knowledge acquisition processes, the sources of information at each level would be helpful.

Response

Thank you for this important comment. We have now presented a summary table in the results section which depicts how the factors at the various levels of the SEM interact to influence the knowledge acquisition process among VYAs. Please refer to line number 354 to 355

23. Line 631-633 “The study has several strengths. Firstly, the use of the SEM provided us with a framework for understanding how multiple and interacting factors affected the SH knowledge acquisition processes among the VYA girls.”

I don’t see the multiple and interacting factors that affected the SH knowledge acquisition process?

Response

Thank you for this important comment. In addressing the comment above, we have revised the themes and presented them in 3 categories according to the SEM. We trust that the revision we have done and Table 3 show more clearly the multiple and interacting factors at the different levels of the SEM and how these influence knowledge acquisition process among VYAs. Please refer to the results section from line number 365 to 600 and to Table 3.

24. Was it possible to conduct a comparison between the VYAs from the implementation sites and VYAs outside the implementation areas? Was there a difference in a source (apart from the clubs, did they communicate better with their parents and peers), was there a difference in the knowledge acquisition process, factors influencing the knowledge acquisition process, was the quality of knowledge different? Line 197 that themes were compared between the two groups. Could you present them?

Response

Thank you for these important questions. We used a constant comparative analysis approach to compare the GOC participants and Non-GOC participants. We have added this explanation in the data analysis section line 264 to 272. The themes that we compared are listed in lines 264 to 267. In the results section line 582 to 643 we have presented the differences in relation to the sources of SH information and knowledge acquisition process, the accuracy or correctness of knowledge obtained, between club participants and Non-GOC participants. 

25. What role did media and technology play if explored? Do the VYAs have access to media and technology, including mass media (radio and TV), social media etc..?

Response

Thank you for this important question. We did not explore access to mass media and technology in this study. We have now acknowledged this as a limitation and provided a recommendation for future studies about the importance of exploring the role of media and technology as a source of SH for VYAs. Please refer to line number 768 to 771

26. Minor comment

1. Your abstract is unstructured and a little bit difficult to follow. Can you provide a structured abstract and especially emphasis on the themes and the key findings from your analysis

Response

Thank you for this important comment. For this manuscript, we used an unstructured abstract format because it is the journal requirement for PLOS ONE. 

27. Can you provide the semi-structured interview guide as a supplementary material?

Response

Thank you for this important request. We have provided the semi-structured guides used in the study as supplementary material.

---

## [Decision Letter · Decision Letter 1]

14 Nov 2023

Sexual health knowledge acquisition processes among very young adolescent girls in rural Malawi: Implications for sexual and reproductive health programs

PONE-D-22-27569R1

Dear Dr. Manda,

We’re pleased to inform you that your manuscript has been judged scientifically suitable for publication and will be formally accepted for publication once it meets all outstanding technical requirements.

Kind regards,

Catherine Aicken

Guest Editor

PLOS ONE

Additional Editor Comments (optional):

I apologise for the amount of time it has taken to provide a decision. This has been due to the reviewer's other academic commitments and my own sick leave.

You will note that reviewer 1, in response to question 6 below, has provided a comment. I am fine with approving the manuscript for publication as it is - i.e. without further changes - however you may wish to take on board the reviewer's constructive feedback in your future qualitative research.

Finally, as someone with a personal interest in Malawian health promotion and public health research (I spent my own adolescence living in Malawi), I wanted to thank you / zikomo kwambiri for the opportunity to handle your interesting manuscript.

Reviewers' comments:

Reviewer's Responses to Questions

**Comments to the Author**

1. If the authors have adequately addressed your comments raised in a previous round of review and you feel that this manuscript is now acceptable for publication, you may indicate that here to bypass the “Comments to the Author” section, enter your conflict of interest statement in the “Confidential to Editor” section, and submit your "Accept" recommendation.

Reviewer #1: All comments have been addressed

2. Is the manuscript technically sound, and do the data support the conclusions?

Reviewer #1: Yes

3. Has the statistical analysis been performed appropriately and rigorously? 

Reviewer #1: N/A

4. Have the authors made all data underlying the findings in their manuscript fully available?

Reviewer #1: Yes

5. Is the manuscript presented in an intelligible fashion and written in standard English?

Reviewer #1: Yes

6. Review Comments to the Author

Reviewer #1: The authors have responded satisfactorily to all my queries and the manuscript has improved substantially. However, I am still a little concerned the results are still a little too descriptive.

7. PLOS authors have the option to publish the peer review history of their article (what does this mean?). If published, this will include your full peer review and any attached files.

Reviewer #1: No

---

## [Editor Report · Acceptance letter]

14 Feb 2024

PONE-D-22-27569R1 

PLOS ONE

Dear Dr. Chimwaza-Manda, 

I'm pleased to inform you that your manuscript has been deemed suitable for publication in PLOS ONE. Congratulations! Your manuscript is now being handed over to our production team.

Kind regards, 

on behalf of

Dr. Catherine Aicken 

Guest Editor

PLOS ONE